**:PLOS | ONE**

# Pulmonary involvement in human visceral leishmaniasis: Clinical and tomographic evaluation

**Ana Jovina Barreto Bispo**[1☯]*, **Maria Luiza Dória Almeida**[1,2☯¤], **Roque Pacheco de Almeida**[1,2☯¤], **José Bispo Neto**[3‡], **Allan Valadão de Oliveira Brito**[3‡], **Camila Mendonça França**[2‡]

1 Postgraduate Program in Health Sciences, Federal University of Sergipe, Aracaju, Sergipe, Brazil,
2 Medical College of the Federal University of Sergipe, Department of Medicine, Aracaju, Sergipe, Brazil,
3 Radiology Service, University Hospital, Federal University of Sergipe, Aracaju, Sergipe, Brazil

☯ These authors contributed equally to this work.
¤ Current address: Postgraduate Program in Health Sciences, Federal University of Sergipe, Aracaju, Sergipe, Brazil
‡ These authors also contributed equally to this work.
* anajovina70@gmail.com

**Data Availability Statement:** All relevant data are within the manuscript and its Supporting Information files.

## Abstract

Visceral leishmaniasis (VL) is a severe, systemic and potentially lethal parasitosis. The lung, like any other organ, can be affected in VL, and interstitial pneumonitis has been described in past decades. This research aimed to bring more recent knowledge about respiratory impairment in VL, characterizing pulmonary involvement through clinical, radiographic and tomographic evaluation. This is an observational, cross-sectional study that underwent clinical evaluation, radiography and high-resolution computed tomography of the chest in patients admitted with the diagnosis of VL in a university service in Northeast Brazil, from January 2015 to July 2018. The sample consisted of 42 patients. Computed tomography was considered abnormal in 59% of patients. Images compatible with pulmonary interstitial involvement were predominant (50%). The most observed respiratory symptom was cough (33.3%), followed by tachypnea (14.1%). Chest radiography was altered in only four patients. VL is a disease characterized by systemic involvement and broad spectrum of clinical manifestations. The respiratory symptoms and tomographic alterations found show that the involvement of respiratory system in VL deserves attention because it is more common than previously thought. Chest X-ray may not reveal this impairment.

## Introduction

Visceral leishmaniasis (VL) is a severe systemic infection caused by sandfly-borne trypanosomatous protozoa that has the dog as main reservoir in urban areas [1]. Globally distributed, it occurs in Asia, Europe, the Middle East, Africa and the Americas, with Brazil accounting for approximately 90% of all cases on the American continent [1–2]. Considered a neglected disease, it is among the six priority endemics in the world (TDR / WHO) [3].

**Funding:** The authors received no specific funding for this work.

**Competing interests:** The authors have declared that no competing interests exist.

It is a systemic spectral disease whose clinical presentation varies from discrete and oligo-symptomatic to moderate and severe forms that can progress to death in 90% of cases if left untreated [1,2,4]. The most affected organs in VL are bone marrow, spleen and liver [5], justifying the classic parasitic condition, characterized by the presence of prolonged fever, hepatosplenomegaly, anemia [1–5].

Cough is a common symptom in VL patients, both in the oligosymptomatic form [6–8] and in the classical form of the disease [9–10]. Respiratory symptoms at admission have been related to worse prognosis in children, possibly due to existing pulmonary infection [11]. However, pulmonary involvement is poorly described. The alteration already reported is interstitial pneumonitis [12–13] and bronchopneumonia is often described as a complication and cause of death in patients with VL [9, 14–15]. There are few references on imaging diagnosis of lung lesions [16].

Due to the high severity potential of VL, it is of fundamental importance to identify patients who are more likely to evolve to situations of greater severity and death, in order to adopt appropriate prophylactic and therapeutic actions and reduce lethality. The present study aimed to describe the symptoms and pulmonary radiology, to associate respiratory symptoms with tomographic alterations. Better knowledge of the disease will help prevent respiratory complications and reduce mortality, as well as minimize the possible chronic effects of visceral leishmaniasis.

## Materials and methods

We conducted a descriptive study with a cross-sectional design not compared with inpatients at the University Hospital of the Federal University of Sergipe (HU-UFS) located in the municipality of Aracaju, capital of the state of Sergipe, in the Northeast of Brazil. A non-probabilistic sample of inpatients with confirmed diagnosis of VL from January 2015 to July 2018 was used.

Patients of any age which were admitted to the pediatric and infectious wards during the study period and had a confirmed diagnosis of VL were included. A confirmed diagnosis was the finding of the parasite in bone marrow aspirate, the positive immunochromatographic rapid test or reactive immunofluorescence reaction with a titer of 1:80 or higher. Patients transferred from other health units in the region who had already started specific treatment for the parasitosis at the time of admission to the study hospital were excluded.

The patients and/or guardians were approached at the HU-UFS ward and were explained about the research and its methods. After agreeing to participate and signing the free and informed consent form, all patients underwent clinical and epidemiological evaluation, physical examination and anthropometric evaluation, with subsequent completion of an individual evaluation form prepared by the study researcher. Subsequently, the patients were referred for chest radiography and high-resolution computed tomography.

Simple chest X-ray was performed at the radiology department of the HU-UFS, in the posteroanterior and lateral views, at maximum inspiration and with the upright chest within 72 hours of admission. The reports were issued by radiologists of the service.

Tomography was performed according to radiology service availability before or at most within 48 hours of treatment initiation. All examinations were performed without contrast on a Toshiba aquillion 64-channel CT scanner, scanning the entire chest caudocranially for complete inspiration. Cuts were made with 1.0mm thickness, 1.5s time interval. For image reconstruction, a high resolution 512x512 dot matrix algorithm was used. The reports were issued by radiologists of the HU-UFS and later reviewed by another radiologist, also belonging to the hospital staff. In discordant reports, the reviewer radiologist's diagnosis was considered.

An exploratory analysis of the data was performed. Categorical variables were described by absolute and relative frequency. Associations between categorical variables were tested using Pearson's chi-square test with Monte-Carlo and Fisher's exact simulations. Continuous variables were described as mean, standard deviation, minimum and maximum. Adherence to normal distribution was tested by the Shapiro-Wilks test. As this adherence was not verified in any variable, the Mann-Whitney test was used to test the median differences. The significance level adopted was 5% and the software used was the R Core Team 2019.

The project was submitted to the Universidade Federal de Sergipe Research Ethics Committee, approved under CAE 14521913.3.0000.5546.

## Results

### Sample characterization

Forty-two patients participated in the study. There was a predominance of males (66.6%). It is noteworthy that in some age groups no female patient was observed.

The average age was 19.5 years. Half of the patients were up to 10 years old; Children up to 5 years old represented 28% of the total patients, about 25% of the individuals were aged between 30 and 50 years and only 10% of the patients were over 50 years old.

In 40% of patients, the time elapsed between symptom onset and hospitalization was between 30 and 60 days. The shortest time between the onset of the first symptom and the moment of hospital admission was 5 days, while the maximum was 180 days.

Serological screening for human immunodeficiency virus (HIV) infection was requested for all patients upon admission. HIV / Leishmania coinfection was detected in two patients.

Another patient with VL had co-infection with schistosomiasis mansoni. No other previous infections were reported by the patient or diagnosed during hospitalization. Four patients had previous diseases, namely: diabetes mellitus (one case), diabetes mellitus associated with systemic arterial hypertension (one case), hypothyroidism (one case), hypothyroidism associated with systemic arterial hypertension (one case).

### Disease characterization

The rk39 rapid test was performed on all patients, with 97% positivity (41/42). The positivity of the myelogram was 64.7% (11/17) and the indirect immunofluorescence was 72.7% (8/11).

The frequency of respiratory manifestations was 49.7%. The most observed respiratory symptom was cough (33.3%), referred to as dry by eight patients (19%) and as productive by six (14.1%). The second respiratory symptom was tachypnea (14.1%), which was always associated with cough. Chest pain was seen in one patient (2.3%) in association with cough and tachypnea.

The general manifestations observed, in order of frequency, were splenomegaly, fever, hepatomegaly, and skin and mucosa pallor.

Results of hemoglobin, neutrophils, platelets, albumin and creatinine can be seen in Table 1.

Specific treatment was performed with glucantime in 21 patients, amphotericin B in 17 and four started treatment with glucantime and ended with amphotericin B. at increased risk of death, according Ministry of Health to clinical and laboratory criteria, two patients. The outcome observed in all cases was medical discharge.

### Chest X-ray

Alterations on simple chest radiography were found in only five patients (Table 2).

**Table 1. Laboratory tests of patients admitted with VL at HU-UFS, from January 2015 to July 2018.**

| LABORATORY TEST | Mean (± SD) | Maximum | Minimum |
|---|---|---|---|
| Hemoglobin (g/dl) | 8,67 (±1,41) | 11,60 | 4,38 |
| Neutrophils (cells/mm3) | 1.019,55 (±1.002) | 4.150 | 147 |
| Platelets (cells/mm3) | 107.264 (±58.648) | 287.000 | 16.000 |
| Albumin (g/dl) | 2,93 (±0,72) | 4,50 | 0,72 |
| Creatinine (mg/dl) | 0,62 (±0,30) | 1,50 | 0,2 |

SD = standard deviation.

**Table 2. Chest X-ray results of patients admitted with VL at HU-UFS, from January 2015 to July 2018.**

| CHEST X-RAY | N (%) |
|---|---|
| **Changed** | 5 (11,9) |
| Paracardiac alveolar condensation with air bronchogram | 1 (2,3) |
| Condensation in the middle lobe | 1 (2.3) |
| Air bronchogram | 1 (2,3) |
| Bilateral pleural effusion | 1 (2,3) |
| Dense streaks on right base | 1 (2,3) |
| **Normal** | 37 (88) |

N—absolute frequency. %—relative frequency.

## High resolution computed tomography

Tomographic changes were found in 59% of patients. The most frequently observed changes were ground-glass opacities, reticular opacities, pleural effusion, and alveolar opacities. The description of all tomographic findings can be seen in Table 3.

Ground-glass opacities were associated with peribronchovascular thickening, bronchiectasis, bronchioloectasis, pleural effusion in one patient.

**Table 3. High resolution computed tomography results of patients admitted with VL at HU-UFS, from January 2015 to July 2018.**

| HRCT | N (42) | % |
|---|---|---|
| Reticular opacities | 8 | 19,0 |
| Ground-glass opacities | 7 | 16,6 |
| Pleural effusion | 5 | 11,9 |
| Alveolar opacities | 3 | 7,1 |
| Bronchiectasis e bronchioloectasis | 2 | 4,7 |
| Peribroncovascular interstitial thickening with micro nodular aspect | 2 | 4,7 |
| Tree-in-bud opacities | 2 | 4,7 |
| Atelectasis | 2 | 4,7 |
| Thickening pleural | 2 | 4,7 |
| Diffuse interstitial infiltrate | 1 | 2,3 |
| Increased parenchymal attenuation | 1 | 2,3 |
| Dense parenchymal striaes and para-septal emphysema | 1 | 2,3 |

N—absolute frequency. %—relative frequency.

**Table 6. Association between tomographic changes and laboratory tests of patients admitted with VL at HU-UFS, from January 2015 to July 2018.**

| LABORATORY TESTS | HRCT | | |
|---|---|---|---|
| | Normal – N (%) | Changed – N (%) | p-value |
| **Anemia** | | | 1,000 F |
| Severe anemia | 1 (6,3) | 3 (11,5) | |
| Moderate anemia | 15 (93,8) | 23 (88,5) | |
| Mild anemia | 0 (0,0) | 0 (0,0) | |
| **Neutropenia** | | | 0,960 QM |
| Severe neutropenia | 6 (37,5) | 8 (30,8) | |
| Moderate neutropenia | 6 (37,5) | 11 (42,3) | |
| Mild neutropenia | 1 (6,2) | 2 (7,7) | |
| Without neutropenia | 3 (18,8) | 5 (19,2) | |
| **Thrombocytopenia** | | | 0,093 QM |
| Severe thrombocytopenia | 2 (12,5) | 7 (26,9) | |
| Moderate thrombocytopenia | 6 (37,5) | 4 (38,5) | |
| Mild thrombocytopenia | 8 (50,0) | 10 (15,4) | |
| Without thrombocytopenia | 0 (0,0) | 5 (19,2) | |
| **Albumin** | | | 1,000 F |
| >3,0 g/dl | 3 (18,8) | 6 (23,1) | |
| ≤3,0 g/dl | 13 (81,3) | 20 (76,9) | |
| **Creatinine** | | | 1,000 F |
| <1,2 mg/dl | 16 (100,0) | 25 (96,2) | |
| 1,2-2,0 mg/dl | 0 (0,0) | 1 (3,8) | |
| >2,0 mg/dl | 0 (0,0) | 0 (0,0) | |

N - absolute frequency. % - relative frequency. F- Fisher's Exact Test. QM - Qui-quadrado de Pearson's Test.

**Fig 1. A 29-year-old male with consolidations and tree-in-bud opacities in the left lower lobe, ground-glass opacities right lower lobe, splenomegaly and hepatomegaly.**

In a second patient ground-glass opacities were associated with consolidations and tree-in-bud opacities (Fig 1); with fissural thickening and pleural effusion in a third; with atelectasia in one fourth patient and as isolated HRCT finding in three patients.

Tree-in-bud opacities was also found associated peribronchovascular interstitial thickening (Fig 2).

Reticular opacities were seen associated with pleural effusion in one patient; ground-glass opacity in one second and atelectasis in a third. In five other patients it was the only tomographic alteration found. Pleural effusion was seen associated with interstitial infiltrate,

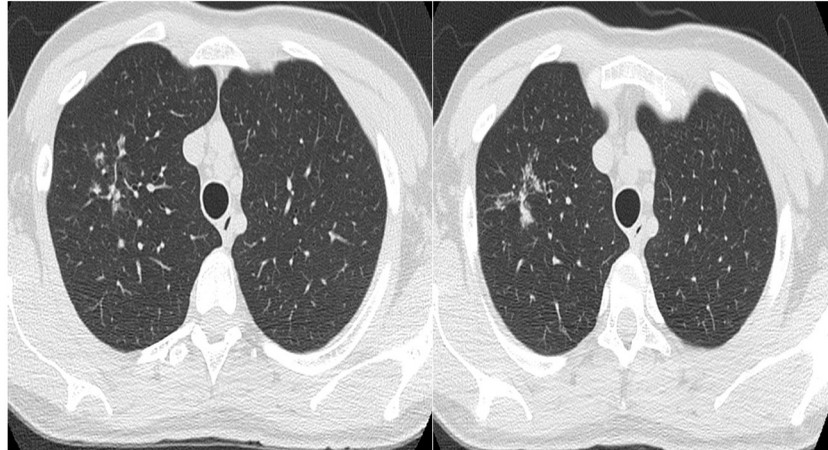

**Fig 2. A 45-year-old male with peribronchovascular interstitial thickening and opacities with coalescent micronodular aspect in apical segment of the right upper.**

**Table 4. Association between tomographic changes and clinical manifestations of patients admitted with VL at HU-UFS, from January 2015 to July 2018.**

| CLINICAL MANIFESTATIONS | HRCT | | |
|---|---|---|---|
| | Normal—N (%) | Changed—N (%) | p-value |
| **Dry cough** | 1 (5) | 7 (31,8) | **0,047** [F] |
| **Productive cough** | 3 (15) | 3 (13,6) | 1,000 [F] |
| **Diarrhea** | 2 (10) | 7 (31,8) | 0,135 [F] |
| **Asthenia** | 11 (55) | 15 (68,3) | 0,527 [F] |
| **Weight lost** | 7 (35) | 9 (40,9) | 0,758 [F] |

N—absolute frequency. %—relative frequency.

[F]—Fisher's Exact Test.

reticular opacity and as a unique finding. Alveolar opacities, increased parenchymal attenuation, dense parenchymal striae, and para-septal emphysema were isolated findings not associated with other changes.

One patient coinfected with HIV and another with Schistosoma had normal HRCT. In the second patient with HIV, pleural thickening and ground-glass opacity were found.

The time interval between symptom onset and hospitalization differed between patients with and without changes HRCT. In patients who had HRCT without pathological findings, the average time between onset of the first symptom and hospital admission was 33.9 days (SD±27.7), while in those who had altered HRCT, this time was 58 days. (SD ±57.2).

There was a statistically significant association between dry cough and HRCT with changes (Table 4).

Physical finding studied were not associated with tomographic changes (Table 5). The association was also not seen with biochemical alterations, as can be seen in Table 6.

## Discussion

VL is a disease characterized by systemic involvement and a broad spectrum of clinical manifestations. Pulmonary involvement in this parasitosis is even considered uncommon in immunocompetent patients [17]. However, the frequency of respiratory symptoms and tomographic changes found in the present study demonstrate that the involvement of the respiratory system in VL is more common than previously thought and deserves attention.

In the late 1990s Cerri *et al* conducted a study using HRCT to prove the interstitial involvement suggested by Duarte, Matta and Corbett ten years earlier. When studying autopsy pulmonary fragments of patients with VL [12,13,16]. The radiological study described reticular

**Table 5. Association between tomographic changes and physical finding of patients admitted with VL at HU-UFS, from January 2015 to July 2018.**

| PHYSICAL FINDING | HRCT | | |
|---|---|---|---|
| | Normal—N (%) | Changed—N (%) | p-value |
| **Splenomegaly** | 10 (100) | 21 (95,5) | 1,000 [F] |
| **Hepatomegaly** | 18 (90) | 18 (81,8) | 0,665 [F] |
| **Pallor** | 16 (84,2) | 18 (81,8) | 1,000 [F] |
| **Tachypnea** | 2 (10) | 4 (18,2) | 0,665 [F] |
| **Edema** | 1 (5) | 3 (13,6) | 0,608 [F] |

N—absolute frequency. %—relative frequency.

[F]—Fisher's Exact Test.

**Table 6. Association between tomographic changes and laboratory tests of patients admitted with VL at HU-UFS, from January 2015 to July 2018.**

| LABORATORY TESTS | HRCT | | p-value |
|---|---|---|---|
| | Normal—N (%) | Changed—N (%) | |
| **Anemia** | | | 1,000 [F] |
| Severe anemia | 1 (6,3) | 3 (11,5) | |
| Moderate anemia | 15 (93,8) | 23 (88,5) | |
| Mild anemia | 0 (0,0) | 0 (0,0) | |
| **Neutropenia** | | | 0,960 [QM] |
| Severe neutropenia | 6 (37,5) | 8 (30,8) | |
| Moderate neutropenia | 6 (37,5) | 11 (42,3) | |
| Mild neutropenia | 1 (6,2) | 2 (7,7) | |
| Without neutropenia | 3 (18,8) | 5 (19,2) | |
| **Thrombocytopenia** | | | 0,093 [QM] |
| Severe thrombocytopenia | 2 (12,5) | 7 (26,9) | |
| Moderate thrombocytopenia | 6 (37,5) | 4 (38,5) | |
| Mild thrombocytopenia | 8 (50,0) | 10 (15,4) | |
| Without thrombocytopenia | 0 (0,0) | 5 (19,2) | |
| **Albumin** | | | 1,000 [F] |
| >3,0 g/dl | 3 (18,8) | 6 (23,1) | |
| ≤3,0 g/dl | 13 (81,3) | 20 (76,9) | |
| **Creatinine** | | | 1,000 [F] |
| <1,2 mg/dl | 16 (100,0) | 25 (96,2) | |
| 1,2–2,0 mg/dl | 0 (0,0) | 1 (3,8) | |
| >2,0 mg/dl | 0 (0,0) | 0 (0,0) | |

N—absolute frequency. %—relative frequency.

[F]—Fisher's Exact Test.

[QM]—Qui-quadrado de Pearson's Test.

opacities, ground-glass opacities, and inter- and intra-lobular septal thickening alterations indicative of interstitial pneumonia [18, 19]. The results of our study, besides confirming, revealed the high frequency of interstitial involvement in the parasitosis. Furthermore, it showed the presence of HRCT findings not demonstrated on chest radiography, such as pleural effusion, parenchymal consolidations, atelectasis, bronchiectasis. Similar results were seen in an imaging study of chronic pulmonary schistosomiasis, which demonstrated that HRCT may show subtle imaging findings lost on plain radiography [20].

The lack of description in the tomographic reports of subpleural lesions may suggest the diagnosis of unspecified interstitial pneumonia, a feature that distinguishes it from usual interstitial pneumonia [21,22].

Frosted glass is a finding considered by many authors to correspond to a pre-fibrotic stage of moderate severity. Even though the disease evolution time was longer in patients with tomographic alteration, the honeycombing pattern was not found, considered the most advanced stage of interstitial disease [18,19]. Studies in patients with systemic sclerosis have revealed that this pre-fibrotic stage is potentially reversible when appropriate treatment of underlying disease is instituted [23]. We consider that, because VL is an acute and treatable disease, the interstitial pneumonia found is a reversible disease. However, patient follow-up may need to be confirmed.

The normality of chest X-rays in the vast majority of patients confirms that this test should not be considered a first choice for interstitial impairment and that HRCT is more sensitive

for the diagnosis of pre-fibrotic and potentially treatable early stages of inflammatory bowel disease [23].

The predominant respiratory symptom was cough (47%), both dry (19%) and productive (14.1%). The frequency of cough found was very close to that reported by Lima in a review of medical records of children who were hospitalized from January 2010 to December 2012 at HU-UFS [24]. Georgiadou *et al.* (2015), in a study conducted in Greece described that cough was the fifth most frequent symptom in VL patients [25]. Queiróz in a study conducted in Recife found a frequency of 42.6% of cough [10]. Badaró (1986), in a study in Northeastern Brazil, had already shown cough as a frequent symptom also in oligosymptomatic patients, only after diarrhea [7].

Tachypnea, present in 14.1% of patients, was the second most common respiratory symptom and was always associated with cough. It is noteworthy that this manifestation is usually described in immunocompetent patients only as presentation of unusual cases of VL [26–28].

Chest pain is a rarely described symptom in patients with VL. In this investigation, chest pain was the least mentioned respiratory symptom, occurring in only one patient, associated with cough. This patient had normal chest radiography and alveolar consolidation TACR. Dasgupta (2017), in a case report from India, described chest pain as an initial manifestation of VL, also associated with cough, but in a patient with pleural effusion [17].

Respiratory symptomatology in LV is often associated with infecctions [10,29]. Diamantino (2010) described pneumonia characterized by tachypnea-associated fever and cough in 12.8% of VL patients, with inference of chest X-ray abnormalities in all symptomatic respiratory patients [30]. Chest X-ray revealed pneumonia in only two patients in the study, and both reported respiratory symptoms. Nodular tomographic pattern and increased pulmonary opacity found in our sample are considered suggestive of interstitial disease, but may also be due to bacterial infection [17,31]. Thus, the frequency of secondary bacterial pneumonia may be higher than that found on plain radiographs. It is noteworthy that on chest X-ray, attention cannot be focused only on the infectious process, but also on the search for changes suggestive of interstitial injury, which are peripheral reticular nodular infiltrates and reduced lung volume.

Pulmonary involvement is known in patients co-infected with Leishmania and HIV [26]. In these patients, there is usually no impairment of the pulmonary interstitium, but atypical findings, such as solitary pulmonary nodule, pleural effusions, mediastinal adenomegalies. Thus, ground-glass opacity found in a patient with HIV must be due to leishmania infection.

We sought to observe if the pulmonary impairment found could be associated with biochemical alterations such as hypoalbuminemia, anemia, neutropenia, thrombocytopenia or creatinine elevation. Such an association was not found.

The most commonly reported general clinical manifestations in our study, namely splenomegaly, hepatomegaly, fever, pallor, correspond to the classic signs of the disease [24, 32, 33].

The diagnosis of VL may be based on clinical and epidemiological data, parasitological methods, immunodiagnostic methods and molecular methods [1,2]. The most commonly used diagnostic method in this study was the rK39 rapid test, a noninvasive and economical test considered highly sensitive and specific in Brazil and the Indian subcontinent [34]. The series of Xavier-Gomes *et al.* (2009) obtained 100% positivity in this exam, while in our sample, its positivity was 97% [35].

## Conclusions

The present study was a pioneer in the joint investigation of respiratory symptomatology, tomographic alterations. The cross-sectional design did not allow establishing causal

inferences between the studied variables, which constitutes a limitation of the research. This was the design chosen because we did not consider it prudent to subject healthy patients to a CT scan to have a control group. Our results were relevant for pioneering, but above all, for bringing recent and objective knowledge about respiratory involvement in a highly incident, potentially serious and lethal disease.

Our findings recommend that patients with respiratory symptomatology undergo HRCT, since conventional chest radiography cannot detect the interstitial pulmonary impairment in VL. Transthoracic ultrasonography, a diagnostic method recently considered useful in the diagnosis of interstitial diseases, could also be an alternative for this evaluation. Spirometry, a low complexity test directed to the respiratory functional study, could also be inserted in the routine of patients with VL, as well as in outpatient follow-up, since pulmonary interstitial impairment may result in restrictive functional alteration.

## Supporting information

**S1 Fig. A 29-year-old male with consolidations and tree-in-bud opacities in the left lower lobe, ground-glass opacities right lower lobe, splenomegaly and hepatomegaly.**
(TIF)

**S2 Fig. A 45-year-old male with tree-in-bud opacities and peribroncovascular interstitial thickening.**
(TIF)

**S1 Table. Laboratory tests of patients admitted with VL at HU-UFS, from January 2015 to July 2018.**
(TIF)

**S2 Table. Chest X-ray results of patients admitted with VL at HU-UFS, from January 2015 to July 2018.**
(TIF)

**S3 Table. High resolution computed tomography results of patients admitted with VL at HU-UFS, from January 2015 to July 2018.**
(TIF)

**S4 Table. Association between tomographic changes and clinical manifestations of patients admitted with VL at HU-UFS, from January 2015 to July 2018.**
(TIF)

**S5 Table. Association between tomographic changes and physical finding of patients admitted with VL at HU-UFS, from January 2015 to July 2018.**
(TIF)

**S6 Table. Association between tomographic changes and laboratory tests of patients admitted with VL at HU-UFS, from January 2015 to July 2018.**
(TIF)

## Acknowledgments

We thank the University Hospital of the Federal University of Sergipe, especially the team of graphic methods and wards.

## Author Contributions

**Conceptualization:** Ana Jovina Barreto Bispo, Maria Luiza Dória Almeida, Roque Pacheco de Almeida.

**Data curation:** Ana Jovina Barreto Bispo, Maria Luiza Dória Almeida, Roque Pacheco de Almeida.

**Formal analysis:** Ana Jovina Barreto Bispo, Maria Luiza Dória Almeida, Roque Pacheco de Almeida, Camila Mendonça França.

**Investigation:** Ana Jovina Barreto Bispo, Maria Luiza Dória Almeida, José Bispo Neto, Allan Valadão de Oliveira Brito, Camila Mendonça França.

**Methodology:** Ana Jovina Barreto Bispo, Maria Luiza Dória Almeida.

**Project administration:** Ana Jovina Barreto Bispo.

**Resources:** Ana Jovina Barreto Bispo.

**Supervision:** Ana Jovina Barreto Bispo, Maria Luiza Dória Almeida, Roque Pacheco de Almeida.

**Validation:** Ana Jovina Barreto Bispo.

**Visualization:** Ana Jovina Barreto Bispo.

**Writing – original draft:** Ana Jovina Barreto Bispo.

**Writing – review & editing:** Ana Jovina Barreto Bispo, Maria Luiza Dória Almeida, Roque Pacheco de Almeida, José Bispo Neto, Allan Valadão de Oliveira Brito.

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
