## [Decision Letter · Decision Letter 0]

2 Dec 2019

PONE-D-19-23416

Pulmonary Involvement in Human Visceral Leishmaniasis: Clinical and Tomographic Evaluation

PLOS ONE

Dear Mrs. BISPO,

Thank you very much for submitting your manuscript "Pulmonary Involvement in Human Visceral Leishmaniasis: Clinical and Tomographic Evaluation" (#PONE-D-19-23416) for review by PLOS ONE. As with all papers submitted to the journal, your manuscript was fully evaluated by academic editor (myself) and by independent peer reviewers. The reviewers appreciated the attention to an important health topic, but they raised substantial concerns about the paper that must be addressed before this manuscript can be accurately assessed for meeting the PLOS ONE criteria. Therefore, if you feel these issues can be adequately addressed, we invite you to submit a revised version of the manuscript that addresses the points raised during the review process. We can’t, of course, promise publication at that time.

We would appreciate receiving your revised manuscript by Jan 16 2020 11:59PM. To enhance the reproducibility of your results, we recommend that if applicable you deposit your laboratory protocols in protocols.io, where a protocol can be assigned its own identifier (DOI) such that it can be cited independently in the future. For instructions see: http://journals.plos.org/plosone/s/submission-guidelines#loc-laboratory-protocols

We look forward to receiving your revised manuscript.

Kind regards,

Abdallah M. Samy, PhD

Academic Editor

PLOS ONE

Journal Requirements:

1. Thank you for including your ethics statement:  "The project was submitted to the UFS Research Ethics Committee, approved under CAE 14521913.3.0000.5546.

Patients over 18 years old and those responsible for patients under 18 years old signed the informed consent form, which states: the purpose of the study; the evaluation procedures; the risks and benefits; the voluntariness of the subject's participation; and the responsibility of the evaluator, respecting the privacy and the total reliability of the data.".

a.Please amend your current ethics statement to include the full name of the ethics committee/institutional review board(s) that approved your specific study.

b.Once you have amended this/these statement(s) in the Methods section of the manuscript, please add the same text to the “Ethics Statement” field of the submission form (via “Edit Submission”).

2. Please include your tables as part of your main manuscript and remove the individual files. Please note that supplementary tables (should remain/ be uploaded) as separate "supporting information" files

**Reviewers' comments:**

Reviewer's Responses to Questions

**Comments to the Author**

1. Is the manuscript technically sound, and do the data support the conclusions?

Reviewer #1: Partly

2. Has the statistical analysis been performed appropriately and rigorously? 

Reviewer #1: I Don't Know

3. Have the authors made all data underlying the findings in their manuscript fully available?

Reviewer #1: No

4. Is the manuscript presented in an intelligible fashion and written in standard English?

Reviewer #1: Yes

5. Review Comments to the Author

Reviewer #1: 1. As many co-morbidity / past illness may cause pulmonary changes, how changes of lung in radiology was attributed to Visceral Leishmaniasis? As authors mention in conclusion that “cross-sectional design did not allow establishing causal inferences between the studied variables” – this should be conversed in the discussion.

2. What is the exact number of cases of the present study? According to table 5, total cases classified according to severity of anemia is 42, though the total number of cases classified according to severity plus absence of Neutropenia is 39.

3. Inclusion and exclusion criteria (beside “Patients transferred from other health units who had already started treatment at the unit of Origin”) of the present study must be defined.

4. Clinical features / physical finding should be depicted in a separate table.

5. Rationality of correlating some parameters (for example Thrombocytopenia) with HRCT changes of lung should be explained.

6. As bone marrow examination was not done in every case, rationality of the diagnosis by serological investigations should be discussed.

7. Results of radiological finding of the present study should be compared with other similar studies. Was any additional significant findings obtained in the present study?

8. Some characteristic image of changes in the lung as depicted in CT / X ray should be incorporated.

9. Units of biochemical tests in Table 5 should be mentioned.

10. Appropriate symbol for fractional number should be placed.

11. All reference must be quoted according to instructions to authors of the journal.

6. PLOS authors have the option to publish the peer review history of their article (what does this mean?). If published, this will include your full peer review and any attached files.

Reviewer #1: Yes: Sudipta Chakrabarti

---

## [Author Response · Author response to Decision Letter 0]

5 Jan 2020

Response to Reviewers

Pulmonary Involvement in Human Visceral Leishmaniasis: Clinical and Tomographic Evaluation

Manuscript Number: PONE-D-19-23416

Some additional information / modification of the manuscript required as depicted in comments to authors. 

Comment to Authors

1. As many co-morbidity / past illness may cause pulmonary changes, how changes of lung in radiology was attributed to Visceral Leishmaniasis? As authors mention in conclusion that “cross-sectional design did not allow establishing causal inferences between the studied variables” – this should be conversed in the discussion.

RESPONSE:

We evaluated the presence of previous diseases, co-infections and comorbidities. We found HIV / Leishmania coinfection in two patients and in another schistosomiasis mansoni. No other infections were reported by the patient or diagnosed during hospitalization. Four patients had comorbidities: diabetes mellitus (one case), diabetes mellitus associated with systemic arterial hypertension (one case), hypothyroidism (one case), hypothyroidism associated with systemic arterial hypertension (one case). Tomographic changes were present, regardless of past or associated VL diseases, including in children, adolescents and young adults, which corresponds to about 75% of the sample. Considering the importance of this information, we added in the results and made the necessary considerations in the discussion of our manuscript. We added the consideration about the study design and the establishment of causal inferences in the discussion.

2. What is the exact number of cases of the present study? According to table 5, total cases classified according to severity of anemia is 42, though the total number of cases classified according to severity plus absence of Neutropenia is 39.

RESPONSE: the exact number of cases of the present study is 42. The table was retified. Excuse us for the mistake.

3. Inclusion and exclusion criteria (beside “Patients transferred from other health units who had already started treatment at the unit of Origin”) of the present study must be defined.

RESPONSE: The hospital where the study was conducted is a referral hospital for Leishmaniasis treatment and receives patients from other hospitals. Patients diagnosed with VL who were already on specific treatment when admitted to HU-UFS did not participate in the study. This clarification had been added in the manuscript.

4. Clinical features / physical finding should be depicted in a separate table.

RESPONSE: We performed the separation in different tables.

5. Rationality of correlating some parameters (for example Thrombocytopenia) with HRCT changes of lung should be explained.

RESPONSE: We made that explanation in the discussion

6. As bone marrow examination was not done in every case, rationality of the diagnosis by serological investigations should be discussed. 

RESPONSE: We made that explanation in the discussion

7. Results of radiological finding of the present study should be compared with other similar studies. Was any additional significant findings obtained in the present study?

RESPONSE: There is a similar study in the medical literature conducted in past decades (reference 16), which was cited in the text. We made a more detailed comparison of our study with this one previous study.

8. Some characteristic image of changes in the lung as depicted in CT / X ray should be incorporated.

RESPONSE: We have incorporated images into the manuscript.

9. Units of biochemical tests in Table 5 should be mentioned.

RESPONSE: We added Units of biochemical tests in Table

10. Appropriate symbol for fractional number should be placed.

RESPONSE: We have incorporated Appropriate symbol for fractional number.

11. All reference must be quoted according to instructions to authors of the journal.

RESPONSE: We quoted all reference according to instructions to authors of the journal.

---

## [Editor Report · Decision Letter 1]

9 Jan 2020

Pulmonary Involvement in Human Visceral Leishmaniasis: Clinical and Tomographic Evaluation

PONE-D-19-23416R1

Dear Dr. BISPO,

We are pleased to inform you that your manuscript has been judged scientifically suitable for publication and will be formally accepted for publication once it complies with all outstanding technical requirements.

With kind regards,

Abdallah M. Samy, PhD

Academic Editor

PLOS ONE

---

## [Editor Report · Acceptance letter]

16 Jan 2020

PONE-D-19-23416R1 

Pulmonary Involvement in Human Visceral Leishmaniasis: Clinical and Tomographic Evaluation 

Dear Dr. Bispo:

I am pleased to inform you that your manuscript has been deemed suitable for publication in PLOS ONE. Congratulations! Your manuscript is now with our production department. 

With kind regards,

on behalf of

Dr. Abdallah M. Samy 

Academic Editor

PLOS ONE